# Social Entrepreneurship Opportunities via Distant Socialization and Social Value Creation

Shah Muhammad Kamran [1,*] , Mahvish Kanwal Khaskhely [1], Abdelmohsen A. Nassani [2] , Mohamed Haffar [3] and Muhammad Moinuddin Qazi Abro [1]

1   Institute of Science, Technology and Development, Mehran University, Jamshoro 76062, Pakistan; mahvish.khaskhely@faculty.muet.edu.pk (M.K.K.); moinuddin.abro@faculty.muet.edu.pk (M.M.Q.A.)
2   Department of Management, College of Business Administration, King Saud University, Riyadh 11587, Saudi Arabia; nassani@ksu.edu.sa
3   Department of Management, Birmingham Business School, University of Birmingham, Birmingham B12 0YB, UK; m.haffar@bham.ac.uk
*   Correspondence: kamran.shah@faculty.muet.edu.pk; Tel.: +92-300-3426-886

**Abstract:** Social entrepreneurs are catalysts for social change on account of social value creation and opportunity identification, thereby improving the quality of life. Their contribution to society is particularly significant in times of crises and pandemics. Hence, the world health crisis caused by the COVID-19 pandemic has increased the need for social entrepreneurship across the globe. Despite social entrepreneurship's relevance in social value creation, the studies regarding opportunity identification in times of social distancing are particularly rare. This constructivist-qualitative research fills the gap by employing the *EDraw Mind Map* tool to explore how the COVID-19 pandemic in general, and social distancing in particular, have shaped entrepreneurial opportunities for social innovation. The findings from content analysis reveal that ICT-based services and industry 4.0 hold a promising future during and post-COVID-19 scenario. They can facilitate a transformation of the threats of social distancing into distant socialization benefits and demand fulfillment. Furthermore, the study contributes to providing a comprehensive depiction of the myriad of opportunities created by social entrepreneurs worldwide. It also guides aspiring social entrepreneurs to adopt such technologies and aim for social integration to ensure quality mental health, education, employment, and manufacturing services in low-resource and developing countries' contexts, as they are severely impacted by the pandemic.

**Keywords:** COVID-19 pandemic; social entrepreneurship opportunities; distant socialization; social value creation

## 1. Introduction

COVID-19 pandemic is currently a colossal threat that has jeopardized the social and economic system, meanwhile burdening the fragile health systems across the world. The global community is struggling to respond to the threat according to the available resources and understanding of the hazard. The World Health Organization has already characterized COVID-19 as a pandemic leading to a global public health emergency [1]. The trend and amplitude of the COVID-19 pandemic are unparalleled in recent history; it is estimated that the world will take a decade or more to heal socio-economically from this pandemic [2,3]. To save the international economy and society from the disastrous consequences of the disease, the 20 largest economies (G20) have pledged USD 5 trillion [4].

The pandemic, which started in Wuhan, China, has exposed multifaceted challenges faced by the global communities, ranging from the collapsing health sector and job losses to rising levels of poverty and hunger. As the global economy has been in varying phases of lockdown, those who are hit the hardest are the households who rely on the day-to-day earnings and self-employed individuals who are mostly the laborers and low-level

employees of SMEs, especially in low- and middle-income countries. The financial severity of COVID-19 can be estimated from the figures on special support packages issued by different governments around the world; the US alone issued a whopping USD 2 trillion through the Coronavirus Aid, Relief, and Economic Security Act (CARE Act) on 25 March 2020 [5]. European governments announced economic rescue packages worth a combined USD 1.86 trillion to mitigate the devastation caused by the coronavirus pandemic [6]. India also released a USD 23 billion relief package to help fight the financial hardship of its citizens who are facing adversities [7].

The scope of COVID-19 warranted its classification as a grand societal and economic challenge, calling for a concerted and multi-pronged approach by various disciplines, such as corporations, governments and NGOs, and social entrepreneurship (SE). Thus, the unique circumstances created by COVID-19 are opening windows of opportunities for SE to create social value in times of crisis [8]. Although enterprises are not obliged to work for the common good of society, social enterprises still have a lot to offer for the benefit of humanity given their nature, which is social and compels them to come forward and share the burden with authorities in times of crisis. Likewise, this period of social distancing motivates the entrepreneurs to come up with innovative and feasible ideas to bridge the gap with social innovation for common relief.

This study aims to find out the opportunities for SE in this time of social distancing caused by COVID-19 and how entrepreneurs can change these hardships into opportunities. Since this pandemic is altering the nature of many societal relationships and reliance on social capital networks, the research question of the current study is what is the role of social entrepreneurship in, during, and after pandemics in general, and COVID-19 in particular? Precisely how is SE addressing the issues of global significance, such as disasters and pandemics? One sub-question is, when social distancing is a norm, and usual avenues of interaction and exchange are challenged, how are SEs playing an effective role in solving local issues of global relevance and bringing about social change? This research is a significant contribution to the present critical literature about the importance and dynamic role of SE and opportunities that can be availed during the pandemic. An exhaustive discussion is conducted on the role of SE during a disaster that forces people to stay away from each other and how entrepreneurs fulfill the demand of the market by providing innovative solutions. It provides a relevant theoretical lens and exemplars of how SE transcended the challenges of social distancing and less frequent interactions between the social actors and the consumers to satisfy their needs of healthcare, finance, distance learning and teaching, e-marketing, and e-business.

The structure of the study is as follows; literature was reviewed about the nature, scope, and resilient role of social entrepreneurship in different societies, regions, and countries to establish its importance in times of crisis. Next, focusing on the present scenario, content analysis is carried out to extract the themes and sub-themes of the current activities of SE around the world. In the last section, a detailed discussion is conducted on the themes generated by content analysis, followed by the conclusion, recommendations and limitations, and de-limitations.

## 2. Literature Review

### 2.1. Theoretical Underpinnings of Social Entrepreneurship

Multiple definitions of SE exist in literature, and they come to a consensus on having the social aim as their primary motive [9]. SE are also considered as constantly innovating to achieve their aims. They are termed as "entrepreneurs with a mission" [10], "catalyst for social transformation", and "social problem solvers" [11]. There is a need for SE because of the limited budget at the hands of the government and leaving some of the social facilities to be catered through SE and social innovation [12].

In the entrepreneurship literature, Schumpeter's (1934) conception is frequently characterized by experimenting with new combinations of products and services, improved production and/or delivery processes, exploring untapped markets, sourcing raw material

from more efficient or sustainable channels, or establishing new organizational forms. Similarly, an entrepreneur is a person who is always in search of a change, responds to opportunities, and exploits them accordingly [13]. It complements Kirzner's (1978) theory presented earlier in which he describes entrepreneurship as the activity of being open and alert to changing circumstances and of being able to exploit value-creating opportunities [14]. Although Drucker and Kirzner primarily focused on the commercial type of entrepreneurship, they acknowledged the non-economic and non-commercial aspects of entrepreneurship as well, such as in the field of academia and healthcare. This non-commercial side is termed SE, and theories of Schumpeter, Drucker, and Kirzner have helped in understanding the evolving nature of entrepreneurship. Practically, it can be explained by giving examples of individuals (social activists for women's and others' rights, community organizers, slavery abolitionists, etc.) or organizations (Red Cross, Grameen Bank, among others), which are motivated by social goals regarding the provision of basic necessities, such as food, clothing, and shelter, and meeting other social needs. SE is primarily different from commercial entrepreneurship in terms of motives and agenda, which are social transformation and social change, respectively, instead of sole monetary profit and political or any other power.

### 2.2. Theories of Social Entrepreneurship

The key themes in entrepreneurship can be framed through a myriad of theories, including contingency theory, creation theory, discovery theory, theory of innovation diffusion, and resource dependency theory, among others. However, the traditional theories of entrepreneurship do not suffice in the context of social entrepreneurship. For instance, the resource-based theory typically explains how organizations create a competitive edge in the market. However, it is against the basic premise of SE in terms of giving priority to a social cause and creating societal value instead of garnering higher profitability [15]. Instead, Schumpeterian and Kirznerian theory is better equipped to explain the social entrepreneur's ability to discern the opportunity in uncertain circumstances by embedding the dimensions of instrumental rationality and pro-activeness [16]. Moreover, mobilization and participation theory for social change can also be employed for social entrepreneurship, since this research, in particular, intends to inspire and mobilize the social capital of developed, as well as developing, economies to opt for products, services, and ideas that can mitigate the social, economic, and ecological impact of uncertain circumstances. This is achieved through SE's collective-driven and collective-oriented approach, which relies on collaborations and alliances [17].

### 2.3. Disaster Management through Social Entrepreneurship

Disasters can be naturally occurring, such as tsunamis, famines, landslides, or man-made, such as wars, terrorism, and/or human activity-induced pandemics and toxic chemical fall-outs. They adversely affect people's physical lives in terms of injury, death, destruction, loss of property and physical spaces. Additionally, they hamper the availability of market spaces, such as stores, marts, and places of social connections, such as religious congregation places and parks, restaurants, among others. In Newton's (1997) view, disasters are not isolated phenomena; they rather occur within a social system and are a social phenomenon. This loss of physical and social spaces is mended primarily through the initiatives of SE adopted by commercial entrepreneurs.

Social entrepreneurs make valuable contribution pre-, post-, and during disasters, in acting as a buffer to reduce the shock of the disaster and engage in activism and advocacy on behalf of communities. For instance, pre-disaster activities include raising awareness and providing information to the communities regarding its scale, scope, and probability of occurrence. Additionally, they facilitate the evacuation from the vulnerable areas and settle them in temporary shelters.

During the disasters, they offer relief services, authentic information assessment of current losses, and realistic future projections of the damages people will be facing. After

the disasters, they organize volunteers to search for the missing people, engage in delivering aid and assistance, and lobby for the allocation of government resources for public services in order to facilitate a return to normal life and rebuild their communities. To this end, they also advocate central government to provide them with basic necessities, such as food, shelter, water, and essential healthcare services [18,19].

### 2.4. Paradox of Social Entrepreneurship at the Time of Social Distancing during Pandemic

In December 2019, the world heard of a viral spread of a novel disease within China, its epicenter being Wuhan, which was termed by the scientists as the novel coronavirus (and COVID-19), likely from an animal source. It caused the severe acute respiratory syndrome (commonly abbreviated as SARS-CoV-2). Within two short months, the virus shook the world with its baffling contagious ability causing loss of lives and global panic [20]. By March 2020, the World Health Organization (WHO) reported over 0.4 million positive cases and approximately 20,000 deaths worldwide, the virus spreading its deadly imprint to 197 countries of the world [1]. It is predicted to be one of the deadliest pandemics of the current century, with symptoms such as breathing difficulty, high-grade fever, and dry and persistent cough. The complications include pneumonia, lung, kidney, and heart failure leading to death, and the most vulnerable persons include senior citizens, pregnant women, smokers, and people with chronic illnesses and medical conditions [21]. Additionally, it is observed that the Asian population is more susceptible to this virus as compared to other races of the world [22].

According to the World Health Organization, the only way to contain it is through limiting and/or eliminating physical proximity among people (person-to-person physical contact), and they have termed their varying degrees as quarantine, social distancing, and isolation, respectively, for a certain period. It has become a major health crisis in the world, and the sole precautionary measure against the virus is minimizing human contact to reduce the exponential spread of the disease [20].

Governments around the world initiated public awareness campaigns along with an extended lockdown to increase the likelihood of maximum social distancing. Social distancing, as defined by UNICEF, is avoiding gatherings and maintaining a distance of 6 feet from another person (s). This social distancing is a realistic solution and a boon in terms of preventing the spread of the virus and reducing the burden on the already fragile healthcare systems of the affected countries [23]. However, at the same time, studies report the post-traumatic stress symptoms due to the outbreak of the disease in general, and social distance in particular. Social distancing has not only led to mental issues, irritability, and isolation but also major physical inconveniences, such as inadequate ration and supplies [24,25].

This has opened a major gap and opportunity for stepping-in of SE to ensure meeting general public's, and especially under-privileged people's, physical, emotional, and financial needs.

Even though commercial entrepreneurship is much discussed in the post-disaster academic literature, there is a dearth of critical contributions made by SE during or post-disaster revival and recovery of the communities. Moreover, no discussion is conducted regarding the hurdles that SE must overcome for restoring economic and social health in the communities by both extending their community-based services and introducing new ones to meet the heightened needs (such as food, shelter, financial, emotional, and spiritual support) of those hit by disasters and calamities [19].

Social entrepreneurship is very pertinent in the context of global pandemics, but how SE can be initiated and sustained creatively and efficiently amid the constraints of social distancing to create community benefit and resilience is not yet discussed in the current stream of COVID-19-related studies, which is a gap this study aims to fill.

*2.5. Social Entrepreneurs during Crisis*

SE is not an easy task in the time of disasters and pandemics when it is needed the most. Besides other difficulties, the two serious issues are societal consensus and organizational problems. The societal consensus is a consensus aggregation problem and the challenge of getting members of the community to agree on the types of social goods that should be supplied from a list of demands [26]. The organizational problem is the lack of the required set of resources and skills to address a new and unanswered problem, especially in emergencies [27].

This organizational problem appears as the absence of a business continuity plan, which is crucial for every kind of firm, irrespective of the nature and scale of business. It is as vital for SE as it is for normal firms. Firms typically have predefined plans for business continuity in case of disaster, both pre- and post-disaster, depending on the nature of the business [28]. Entrepreneurship and, more precisely, the social entrepreneurial business continuity plan, emphasize two approaches: either conducting business throughout a disastrous circumstance or reopening afterward, to establish sales streams and profitability by fulfilling the post-disaster demand for social goods. This is achieved by the capabilities of the business to "continue delivery of products or services at acceptable levels following a disruptive incident" [29].

The coronavirus pandemic intensified the demand for SE with social innovations to assist the most vulnerable in society. Today, in the time of COVID-19, SE is needed more than ever because of the negative effect on health and economic downfall of the already marginalized, vulnerable, and those who depend on the day-to-day earnings [30].

One important characteristic at the time of crises is the resilience depicted by enterprises [31], which is their ability to continue operating at the time of crisis and disruption. Conceptually, this ability is fed by the presence of the already accumulated resources in pre-crisis times, which are strategically deployed to ease the impact of the crisis. Similarly, this entrepreneurial resilience is of critical importance during COVID-19. A few studies conducted on crisis management in the entrepreneurship context discuss the steps taken by entrepreneurs to reduce the magnitude of the repercussions, including the adaptation in sales, marketing, human resource management, including employment practices [31]. More specifically, small entrepreneurial enterprises are more flexible and adaptable, and the same is expected of them during COVID-19 and its associated dynamics, such as social distancing and lockdowns [32,33]. Entrepreneurial crisis management is closely associated with the notion of bricolage, which asserts the need for iterative and flexible approaches, including effectual logic instead of adopting rigid practices, especially during a global pandemic such as COVID-19 [34]. Moreover, the resilient entrepreneurs are those who bring positive social change (creating change and opportunities) with their scarce resources, which is an effectual principle [35]. This is precisely a trait that the aforementioned enterprises and initiatives across the globe have exhibited in this hour of need.

## 3. Research Methodology

This research adopts a systematic literature review, followed by a content analysis of electronic sources, culminating in the generation of themes. The detail of the methodology is elaborated in next section.

*Planning the Literature Search for Analysis*

A systematic review process is adopted to ensure replicability for future studies. The systematic review process is considered a reliable and scientific overview of current research on the area of inquiry, while it aims to identify, review, and synthesize all related studies using a transparent and replicable process [35,36]. This review process commences with the setting out of rules and boundaries for exhaustive literature search to analyze and create an ontological classification of the raw data. The principles of precision, transparency, coverage, and thorough synthesis are ensured [37,38].

## 4. Data Analysis

The secondary data is analyzed in two steps. First, systematic literature review is conducted and then, the gleaned themes are generated on the basis of the author's understanding of the SE phenomena under the major headings. The details are discussed in the section below.

### 4.1. Conducting the Literature Search for Analysis

The search for this study involves many steps, and it begins with the protocol. Initial search criteria were adopted in which the conceptual boundary of the term "SE" is drawn through its definition, which involves creating value for the communities by amalgamating, restructuring, and deploying resources in new ways [32]. Secondly, the intention for SE is clear, namely that such innovation, exploration, and exploitation of resources are carried out for the creation of social value and bringing about social change. Further, when the same term is viewed from the perspective of a process, it does not limit itself to offering needed products and services but also includes the conception of new organizations. The guidance for the search term emerged from the literature, and these search terms entered in the pertinent search forums became the decisive criteria for the study's inclusion or exclusion and its relevance for our current research.

Regarding the exclusion criteria, books and book chapters were excluded because COVID-19 is the latest pandemic, and books and chapters addressing its social entrepreneurial aspect are not available due to incoherent peer review process and more limited accessibility, whereas journal articles are assumed to contain validated knowledge [39,40]. However, the major focus remained on the reports and magazine articles because the number of research articles about pandemics, and especially this COVID-19 pandemic, is not enough for a systematic analysis, but sufficient research is available on the significance of SE during and after a disaster and previous pandemics. To address this issue, all published and accessible journal and magazine articles that met the selection criteria were included (Appendix A). This method is helpful for new research ideas at the developmental stage and makes the replication and extensions simple and smooth [37].

Since the scope of this research encompasses the contribution of SE at the time of social crises, pandemics, and disasters, including the latest COVID-19 pandemic, the time-frame of the search is between 2010 and 2021, inclusive of 2010. This starting point is employed, as this study extensively discusses SE's role in the current pandemic, yet it is not limited to COVID-19 and its implications. The research draws parallels with other crises of the past; for instance, the SARS pandemic erupted in 2010, which is similar to the current health crisis, therefore, 2010 met our inclusion criteria. The search was commenced by using key terms and phrases identified from the literature, including "social entrepreneurship", "social distancing and entrepreneurial opportunities", "entrepreneurial resilience during a pandemic", among others. This search generated the pertinent titles from the Google Scholar search engine. All researchers then individually checked the full studies against the admittance and exclusion criteria; to the extent of the search, the resultant list was matched with criteria to find missing items. This extensive process yielded a final categorical data of 103 articles and reports addressing different opportunities for SE during and after disaster/crisis/pandemic. Social enterprises that did not address the entrepreneurial aspects were excluded from the search. Exceptions were made for some articles that discussed the early stages of social enterprises.

Content analysis was performed on the final dataset using the inductive approach to identify the social entrepreneurial opportunities during and after pandemics and disasters. Research design first focused on the prevailing conditions and motivations of the individuals to take initial steps during the time of pandemics. Then, post-disaster and post-pandemic circumstances were examined to ascertain how SE solved social issues in a bid to inform about their initiatives to the aspiring entrepreneurs and motivate them to find such opportunities. This approach is helpful in developing a timeline of how entrepreneurs

can respond to specific situations while thinking about their actions in current and future situations [37].

### 4.2. Conducting the Content Analysis

An inductive approach for theme identification was followed with the application and derivation of ontological and thematic protocol (see Appendix B). Interpretive synthesis was applied as an alternative to the deductive application for a predetermined methodological framework [36]. Therefore, the structure and nature of the domain were decided by the organization, identification, and classification of the study. The themes signify the basic concept of each study; thus, the themes identified in the analysis represent the core ideas, points of view, and theoretical linking of statements on which a study's research problem and propositions are based [38].

Themes were inductively derived from the data (data meaning the studies) by following the principles of thematic coding from qualitative research [39]. The study's objectives, theoretical grounds, and measures developed the understanding of its primary and secondary themes. Themes were extracted from the comprehensive understanding of the articles and reports, in contrast to the conventional content analysis approach that usually extracts themes from de-contextualized information. Thus, the names of the themes were borrowed from the literature.

The theme identification and verification processes were iterative, detailed, and extensive. The first iteration cycle brought a significant number of refined themes; these themes were then sorted out and classified to develop an ontological organization of the scope. The classification process echoes the context of the study, and it is flexible in response to the field [40]. The general principles of ontological design were applied, including the development of a distinct super-class above the sub-class and similar-class, which are vertically arranged after rigorous scrutiny against redundancy and repetition at each level.

Scope ontology ranked 18 themes, by combining similar first-order themes. After sorting them vertically through proper ranking, the second-order themes were compiled (thematic area). This process continued until one key research area of social entrepreneurial opportunities was finalized.

A total of 103 studies were classified into three types: opportunity to assist the government during/after disaster and/or pandemic (Class-I); addressing the social issues during/after disaster and/or pandemic (Class-II); and commercial opportunities during/after disaster and/or pandemic (Class-III). Ontological arrangements of SE opportunities were presented in a thematic map designed using EdrawMindMap 7.5 (Figure 1). The main branches, extending from the left of the map, represent SE's opportunities in disasters and pandemics. These narrowly defined sub-groups further elaborate the types of opportunities during and after pandemics. This cycle leads to the final cumulative total of 103 studies analyzed for the social entrepreneurial opportunities (see Appendix A for a list of studies) (Jones et al. (2011) were followed for research design and methodology).

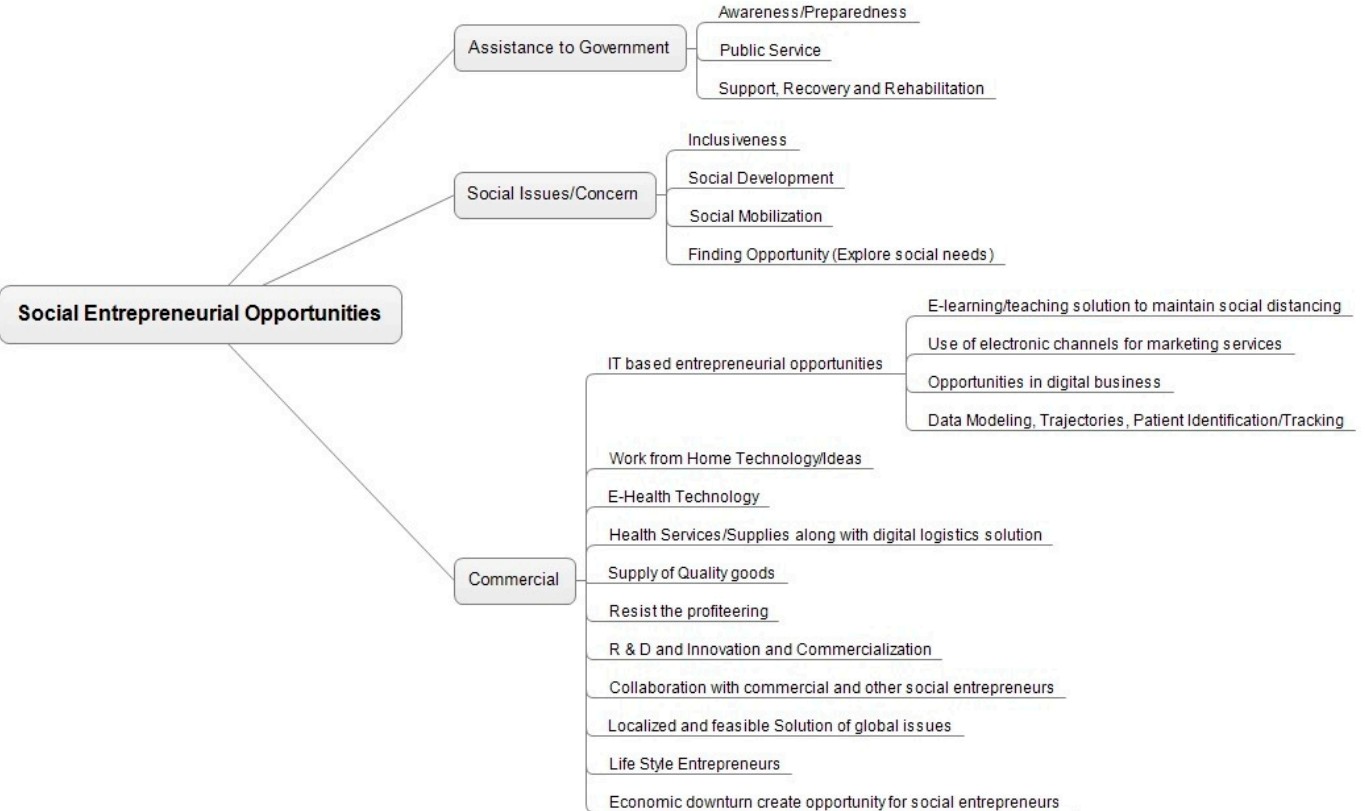

**Figure 1.** Thematic map of social entrepreneurial opportunities.

## 5. Results and Discussion

The thematic map (Figure 1) describes the ontology of SE and visually defines the structure and arrangements for the Results and Discussion section. Results of the analysis show that, broadly, there are three types of opportunities for SE during and after a pandemic/disaster: opportunities to assist the government, addressing the social and societal issues, and commercial opportunities. The map matches the organization of Table 1, which presents details of the themes with descriptions.

**Table 1.** Domain ontology for papers (commercial opportunities for social entrepreneur/ship). Source: author's compilation.

| Second Order Theme | First Order Theme | Theme Description/Explanation |
|---|---|---|
| Assistance to government | Awareness/preparedness | Readiness of the masses |
| | Public service | Supply of public services and goods |
| | Support, recovery, and rehabilitation | Rescue, recovery, and rehabilitation |
| Social issues/concerns | Inclusiveness | Services for marginalized and diverse populations |
| | Social development | Educate about the facts and motivations |
| | Social mobilization | Mobilize the masses for the collective good |
| | Funding opportunity (explore social needs) | Explore the opportunities emerging with time |

**Table 1.** *Cont.*

| Second Order Theme | First Order Theme | | Theme Description/Explanation |
|---|---|---|---|
| | IT-based entrepreneurial opportunities | E-learning/teaching solutions to maintain social distancing | Entrepreneurial opportunities based on ICT |
| | | Use of electronic channels for marketing services | |
| | | Opportunities in digital business | |
| | | Data modeling, trajectories, patient identification/tracking | |
| | Work-from-home technology/ideas | | Remote-working solutions |
| | E-health technology | | Technologies for healthcare and healthcare professionals |
| Commercial | Health services/supplies along with digital logistics solution | | Smart logistic and distribution solutions |
| | Supply of quality goods | | Supply of quality goods when needed the most |
| | Resist the profiteering | | Supply goods and services with reasonable profit |
| | R&D and innovation and commercialization | | Startups in R&D and commercialization of R&D |
| | Collaboration with commercial and other social entrepreneurs | | Strategies for designing collaboration for society |
| | Localized and feasible solutions for global issues | | Localized solutions for local issues |
| | Lifestyle entrepreneurs | | Self-satisfaction and social work |
| | Economic downturn creates opportunity for social entrepreneurs | | Profit with social uplift during rough times |

## 5.1. Entrepreneurial Opportunities

Studies are grouped and tagged as entrepreneurial opportunities, as the articles and reports cover the opportunities available for social entrepreneurs during and after pandemics and disasters. A review of literature finds three thematic areas, as illustrated in Figure 1: (1) assistance to government, (2) social concerns/issues, and (3) commercial opportunities. Details are outlined in Table 1. Next is the analysis of each thematic area.

## 5.2. Assistance to Government

Assistance to government is the first second-order theme that emerged in SE literature. This covers the opportunities SE can find in the public sector by providing services that are primarily the responsibility of government, i.e., recovery and rehabilitation of those affected, awareness about the severity and scale of the pandemic or disaster, and preparedness prior to it hitting. Assistance to the government covers public services separately, i.e., the supply of public services to share the burden with government in times of adversity.

Disaster/pandemic preparedness and awareness appeared as the first, first-order theme. This research contended that small and medium enterprise and established businesses do not have any interest in disaster preparedness and awareness efforts, as it is the legal responsibility of local and national governments to mitigate the risks associated with disasters/pandemics and develop an emergency control strategy [41]. In the event of a disaster or a pandemic, such as COVID-19, governments on any level cannot work alone to address and estimate the needs, educate the people, communicate the possible impact, and effectively disseminate all relevant information to the community. SE can help assist the government in this job. Engagement in preparedness and awareness programs will be beneficial for SE, as well as for the greater social good of society. Social entrepreneurs' involvement in the preparation/awareness process at the initial stage will help to explore new opportunities and consider the particular requirements in the disaster management plan to enhance resilience. At the community's end, the involvement of SE in government disaster management activities will increase the resilience of the respective communities and decrease the disaster's impact. This is because of the localized and socialized na-

ture of entrepreneurship, which binds entrepreneurs to assist the authorities in difficult times [41–43].

The second first-order theme is "public service", discussing SE's contribution to the social welfare system and fulfillment of the promise to bring substantial change in the well-being arena of the community during disasters and pandemics. In the time of disasters, when governments cannot act comprehensively enough to address the multifaceted challenges, except for increasing their expenditure, social entrepreneurs' collaboration with authorities can leverage the resources and offer reframed and efficient solutions [44].

The coronavirus pandemic is revealing many new aspects, particularly the inequalities of national and global socioeconomic system, and governments at any level and of any country cannot solve them alone. The development sector, social enterprises, and SE are needed to share the burden of government and play their part. Studies found that COVID-19 is threatening the livelihood of approximately 1.6 billion workers, nearly half of the global workforce, and economic stimulus packages, even in developed countries, are not reaching the already excluded parts of the society [45].

SE has the potential to resolve the government and market failures by different means, highlighting the issues and areas not in focus, mobilizing the social resources to help the government supply social goods. SE also helps to fulfill the demand for public goods of the most vulnerable and neglected populations, as this section of the community is already vulnerable and at the highest risk of COVID-19, as well as other pandemics and disasters [46–48].

Besides others, SE has the opportunity to provide reliable and authentic information about the pandemic and related issues to the people who have no access to information, and even when they have the required information, they are not able to interpret and use it. SE can also work for the care services for the most neglected and exposed population. These are the responsibilities of government, but due to the budgetary, management, and/or access-related constraints, they are not able to address them [30].

The third first-order theme is "support, recovery, and rehabilitation". Studies in this group focus on the post-disaster and post-pandemic support for recovery and rehabilitation of the community. The literature points out that socioeconomic situations after disasters and pandemics are likely to open up new prospects for SMEs, and especially SE, improving the local and national economy, and also enabling the affected communities to help themselves and rise again [49–51].

Then, in the situation of extremely damaged and diminished resources, SE mobilizes the communities and resources to come together and solve the problems. The entrepreneurs further explain that because of the localized nature of SE, which is based on place and culture, they have the best opportunity to understand and resolve the needs of the community by supplying the desired goods and services, as no-one else will [52].

Previous studies suggest that social entrepreneurs' role in the post-disaster/pandemic recovery should be acknowledged and appreciated because they are pivotal in the assessment and response to the unpredictability associated with these events. Disasters and pandemics often hit hard the most vulnerable communities, and in most cases, after many years of post-recovery efforts, some of the distressed communities remain far from recovery. This is because much of the focus remains on the government's efforts for recovery and rehabilitation, and the government itself needs assistance from the population and communities. It is at this time that SE come forward with a rapid and feasible solution to the prevailing problems [19]. They have the opportunity to promote urban resilience and assist governments with disaster risk management [53].

### 5.3. Social Issues and Concerns

"Social issues and concerns" is the second second-order theme that emerged in SE literature. This theme circles around the social concerns and issues that appear during and after pandemics and disasters, and states that SE has the ability and opportunity to address those issues. Previously, social problems were the sole domain of government or

civil society; today, market mechanisms and approaches are being used to address them. SE is one of them, and it makes a profit while addressing social needs [54].

Inclusiveness is developed as the first first-order theme in social concerns and issues. Studies in this area state that SE can include the marginalized and excluded groups and communities by two means. First, these devastated groups receive the most needed services and goods in time of utter disturbance at all levels due to abnormal situations [55]. Second, integrating the more diverse, ethnic, and religious groups in core activities of entrepreneurship will lead to a wide acceptance of entrepreneurship among various communities and also increase the sense of recognition in the marginalized communities [56].

Studies stressed the need for inclusiveness as a business strategy, not simply an option or CSR. Many startups can adopt an inclusive business model and develop products and services for the population living at bottom of the social pyramid [56]. These research works also propose that engaging a more diverse workforce is an opportunity and also a success factor for SE because it leads to understanding and obtaining more localized and customized solutions to the social problems related to pandemics and disasters [57].

The second and third themes of the first order in "social issues and concerns" are social development and social mobilization, respectively. These studies cover the impact of SE on society and how, in situations of pandemics and disasters, they can play a significant role. Literature related to social mobilization encompasses the social opportunity for entrepreneurs by bringing together all societal and personal influences to raise awareness and demand for solutions to social issues.

Social impact literature demonstrates that SE aim to create social impact and bring about social change along with social transformation. Research in this area emphasizes that pandemics leading to disasters are underlying motivations and prerequisites for the incorporation and growth of SE. Through this, they can create quantifiable social impact [58,59].

SE contribute to transforming the societal system by offering new ideas and approaches to address new, as well as old, issues. The post pandemic/disaster period is crucial for offering a social change to the hard-hit people because they are deprived and in dire need of change. Studies also suggest that SE is much needed in developing countries, where pandemics and disasters make the situation ever worse [60]. However, SE also need other actors to come forward and join hands to make an impact. To activate other actors and stakeholders, SE need to mobilize resources and communities.

A collective approach to address the problems caused by pandemics demands collaborative efforts, cooperation between stockholders and other institutes, and mobilization of the community. These efforts may not bring financial benefits but will bring social change with a mobilization of the community and resources in a constructive manner [61]. A study states that SE has great exposure to social mobilization in terms of resources and community involvement because of the nature of their business, famously called bricolage—a mechanism of change [62]. This bricolage can work as a catalyst in time of pandemics when the community needs help and resources. There is also a need for institutional coordination to minimize the challenges and for optimal performance [63,64].

The last first-order theme in "social issues and concerns" is "funding opportunity". This theme was classified under social issues because the funding for social and commercial startups chiefly focuses on pure social solution that appears during the time of pandemics and social distress. Studies on this theme state that entrepreneurs, and especially SE, are very much aware of the ups and downs of the economic landscape. However, COVID-19 has toppled most of the economy, made thousands of workers unemployed, and affected the investment, which ultimately can impact the funding and financing of startups (Connor, 2020). On the other side of the coin, situations such as pandemics and disasters encourage venture capitalists and governments to fund SE to bring a cure for the disease or a solution to normalize the situation.

Studies have concluded that not only governments and seed financers but large corporations should also support commercial and SE because startups have an agility

edge over established corporations, whereas corporations have resources that startups cannot even dream of. The combination of resources and entrepreneurial activity can boost innovation and creativity [65]. This phenomenon works even better in time of pandemics when most economic activities are on hold, and large corporations start to look for new ideas and marketable products and services, while entrepreneurs search for finance. Besides corporate financing in SE, pandemics and disasters also open the doors to new funding alternatives, e.g., microfinance, peer-to-peer landing, crowdfunding to address social issues and concerns [66].

### 5.4. Commercial Opportunities

"Commercial opportunities" appeared as a second-order theme in the SE opportunity thematic area. They can be further grouped in eleven broader categories (Figure 1). These studies discuss the commercial importance of SE in uncertain time of pandemics and state that entrepreneurs are a source of hope that can help the communities recover. Furthermore, entrepreneurship, either commercial or social, provides the necessary goods and services, establishes and maintains social networks, and, most importantly, brings confidence to the masses that this time of hardship shall pass [67].

The first category in the first-order theme is "IT-based entrepreneurial opportunities", which includes: digital solutions for distance learning and teaching, e-marketing channels, e-business and big data, and AI usage for COVID-19. These studies established that information-technology-based businesses have great potential to serve the community in time of crisis. Furthermore, studies showed that commercial enterprises can use the special interest of youth in IT as the motivation toward social cause and social entrepreneurial initiatives, especially at the time of crisis [68].

Online applications and platforms for distance education aim to help teachers, students, schools, and administration and enable the people to interact, teach, and learn during the periods of social distancing. This crisis has compelled universities and other educational institutes worldwide to explore new ways of reconfiguring the conventional educational programs to make them suitable for distance learning. For instance, an educational entrepreneurial venture called CLab@Salento focuses on technology-centric entrepreneurship opportunities for university students [69]. These applications are designed to make a significant impact through a strong and wide user base. Studies found that a rapid shift to online education has been a success, and the experience will help in developing future strategies, including: new rules, regulations, SOPs, and even laws [70–75].

Studies about e-marketing and e-business emphasize that, as the pandemic continues on for longer, businesses need to be more digital. Situations such as COVID-19 are unprecedented, and nearly all channels related to live events and gathering either socially or commercially are disappearing, along with the enormous challenge of increasing distance in the face-to-face business. One of the possible ways of mitigating this loss of communication is the digitization of marketing channels. This shift could be used in the long run when life returns to normal; it would make companies more buoyant in any future storms of pandemics and disasters [76].

Studies have listed different strategies and opportunities that SE can avail. These opportunities, on the one hand, will help the entrepreneurs start a venture and create jobs; on the other hand, it will be important for the community to return to normal. The most discussed options for digital marketing are blogging and social media posts for SE. These options ensure mass communication, cost effectiveness, and balancing between communication, cause, and activities of SE [77–79].

Similar to the digitization of marketing channels, there appears a separate first-order theme of e-business opportunities during and after disasters and pandemics, along with a specific application, as well as an opportunity based on information technology in big data and AI use for tracing and analyzing COVID-19 patients and the disease itself. Studies suggest that the digital transformation from business to business is a necessity of the time, and companies and business leaders cannot avoid it [80].

Artificial intelligence and big data emerge as potential answers to the coronavirus and other emergencies; their importance will increase in the coming time. These technologies are being used to track the patients and the spread of disease in real time, for developing strategies to impose and lift public health innervations, for effective plans, drug-related discoveries, and repurposing and enhancing the community response to pandemics and disasters [81].

Entrepreneurial startups based on ICT, big data, and artificial intelligence vary in nature and skill level. Some startups, such as Zomato, simply expand their activities from "food-tech to grocery startups to meet surging online-order demand" while others, such as EVQLV, are highly sophisticated and develop algorithms with the ability to generate, screen, and optimize millions of therapeutic antibodies [82]. Studies and reports explored and discussed tech entrepreneurship with a special focus on social needs during and after pandemics leading to disasters [83–91].

Another first-order theme in the area of opportunities emerges from the literature relating to the necessity of the time, which discusses the commercial opportunities posed by the situation for information technology and human resource management. SE are searching for creative solutions to maintain and foster connectedness and community even during the isolation and working from home to maintain social distance. Such activities can provide a sense of community, opportunities for discussion, and ways to maintain and even grow social ties. For instance, during "physical distancing" times, videoconferencing services to conduct a business meeting and happy hours with colleagues are included in work-from-home technologies. Then, there are opportunities to equip people with the means for maintaining the connection through live streaming with communities such as worship houses, community centers, and gyms. Additionally, there are a variety of mobile applications to track health, fitness, and nutrition levels associated with improved user health [92]. This is also true in the case of other industries, as working from home reduces stress due to the time and cost saved on travel and, consequently, an increased satisfaction over time.

For instance, one research work conducted with 604 Irish adults concluded the above-mentioned activities performed during COVID-19 restrictions had positive health effects. So, the technology can provide avenues for administering and enlarging physical and health services while reducing the disparities between rich and poor by including video and teleconferencing interventions [93,94]. Similar work-from-home ideas are discussed in Refs. [95–98]. Therefore, the opportunities provided by the assimilation of IT and healthcare can act as an inflection point to expand their reach to rural and hard-hit areas with low-income populations. Innovation and IT can help control the global pandemic through the mobilization of available resources, help reduce the pressure on the healthcare system, and ensure facilities outreach [99,100].

Studies grouped in the next first-order theme discuss the e-health technology. Technologies for healthcare are also opportunities for healthcare professionals because the unexpected pace of COVID-19 eliminated long-time obstacles to the embracement of change, and it forced the experts and systems to adopt a better and more accessible healthcare system, equipped with inventiveness and technology. The new approach can open many doors for healthcare professionals. Telemedicine and digital therapeutics made it convenient for health professionals to provide proper, on-time, and stay@home health advice to more patients and people in need. Similarly, the mammoth size of a pandemic requires an army of professionals for research and advancement in the field of medicine, so health experts can not only find job opportunities but can also provide consultancy services to the big pharmaceutical companies [100–105].

Following the COVID-19 situation, another theme has emerged in the form of health services/supplies. These studies focus on smart logistics and distribution channels in time of pandemics. These studies conclude that a global pandemic brings opportunities for entrepreneurs in the field of the supply chain of many daily life essentials, including food and healthcare supplies for individuals and organizations. The health and medical

essentials supply chain, in particular, needed to respond against the rapidly increasing demand for its products [106]. A study by an entrepreneurial platform Startus (2020) analyzed the situation and found that entrepreneurs have responded to the demand shock by providing the solution required for a continuous supply of essentials. The study further pointed out that there is a worthwhile opportunity for entrepreneurs in the area of supply chain and logistic solutions for hospitals and, more generally, for everyone. Similarly, the digital logistic solutions provided by university startups for uninterrupted hospital supplies in time of the pandemic are explored, and it is emphasized that the innovators and entrepreneurs come up with feasible and less expensive ideas not only for profit but also for the need of the society. Further, supply chain and logistics-related issues concerning pandemics are also discussed [107].

An important category that appeared in the first-order theme is the supply of quality goods in the difficult and equally needed time of pandemics. The supply chain opportunities for SE have already been discussed above. Here, the quality of goods and especially the quality of personal and protective equipment (PPE) at a reasonable price, is the point of concern. A huge difference between supply and demand for PPE is one side of the problem; the other and more acute side is the quality of the PPE supplied to healthcare professionals [108,109]. Avoiding healthcare workers becoming vectors or patients of the global pandemic relies heavily on the quality of PPE, and its shortage is leading to compromised quality. Considering the importance of quality, as an opportunity, the studies suggest that SE can exploit areas with the dual purpose of social service and profit making at marginal rates [110,111].

Buyers globally have changed their shopping patterns for many reasons, including a price hike. It is further mentioned that shoppers are forced to buy groceries at higher prices they feel are exploitative and repressive [112]. COVID-19 is killing millions of people around the world, but profiteers are using this pandemic for their benefit by collecting government subsidies, polluting the environment, and making life more miserable in these difficult times; the cynicism of these profiteers is creating public health hazards, which adds insult to injury [113]. Even the corporations in "essential" industries, such as healthcare and logistics, are profiteering off the crisis [114,115]. These studies lead to the exploration of another theme of the difference in mindset between commercial and SE.

SE can supply goods and services with a reasonable margin of profit instead of unreasonable price hikes or price gouging, unlike their commercial or conventional counterparts. This difference between reasonable and unreasonable increases could be translated into an opportunity. In pandemics and disasters, when hoarding and supply profiteering prevail [116], an economic downturn creates the space for SE to come forward with their very objective of social well-being. This will not only help them to earn the trust of society, but it will also lead to a recognition by the authorities who are already cracking down against the coronavirus profiteers [117]. Innovative services and goods that meet the pandemic-driven needs, such as Zoom, Teladoc, and Pelaton, can thrive instead of losing.

Further, adverse situations also bring out the best in the area of research and innovation of traditional medicines, unconventional medicine, and so on [118]. They further emphasize that science, technology, and innovation (STI) is the path of progress for the healthcare system and also a bridge to cover the social distance. All that is required is a center for serious and exhaustive research and development. Gupta highlighted the other side of the COVID-19 situation, which is the shortage of labor (construction, agriculture and machine workers, loaders, handlers, drivers) due to social restrictions and lockdown [119]. These two opposite sides of the COVID-19-led situation brought the first-order themes of research, development, innovation, and commercialization of the pandemic in the second-order theme of commercial opportunities.

Studies under this theme point out that even the developing countries transformed this initial disadvantage in the manufacturing, logistics, agricultural, and other arenas to technological innovations, such as Industry 4.0, including the utilization of Internet of Things, robotics, artificial intelligence, virtual and augmented reality (resulting in digital

manufacturing), logistics, and supply chain and green technology for treating waste and improving the quality of public life [119,120]. Hence, these startups in R&D and innovation for social uplift also result in profit generation, and tech startups are making a significant contribution to many sectors of the quarantining economy [121,122].

Research and innovation products remain on the shelf until their commercialization, and it is proven that commercialization of universities, R&D startups, and individual research products can be a source of income for the firms and even for countries, while being an opportunity for some entrepreneurs through sales [123]. Some other studies are also consistent with these and conclude that SE can transfer academia-invented knowledge, technology, and expertise to marketable products and services; therefore, university and education-related entrepreneurs can successfully lay the foundation of startup firms and generate profits through intellectual property while moving innovations from the laboratories to markets and industries for social well-being [124–126].

Besides the above-discussed commercial opportunities for SE, there is another opening for partnership, collaboration, and joint ventures with established businesses, corporations, and other traditional and even social startups. Startups are agile to large corporations, but corporations have more resources than the imagination of startups can comprehend. A partnership between entrepreneurs' agility and the resources of the corporations can result in amazing output [65]. The fight against COVID-19 established a platform for this partnership because large corporations are facing almost the same challenges as entrepreneurs, e.g., unpredictable circumstances, logistical and operational issues due to lockdown around the world, the need to adopt new approaches rapidly. However, a collaboration between the startups can yield more fruitful results in the time of unpredictable circumstances. Collaboration may take different forms, e.g., togetherness, communication, engagement, sharing of resources, and socializing, to name a few [127]. A 2015 study further states that this culture of joint venture and collaboration between startups–startups and startups–corporations can result in innovation and synergy in outputs. For instance, the hospitality industry has suffered tremendously from the pandemic through travel restrictions, canceled flights, and strict global lockdowns, especially the peer-to-peer accommodation category, including Airbnb, where the accommodation cancelation rate was 90% during the peak of the pandemic [66]. However, post-COVID-19, it is speculated that the entrepreneurs will emerge as the driving force of this peer-to-peer accommodation due to their flexibility and swift response to the crisis [128].

Finance and other material resources are the main hurdles for startups, and they can gain access to these resources through collaboration with large corporations, whereas corporations can bring innovation and innovative suppliers in the shape of startups [66, 129,130]. This collaboration is a win–win situation during COVID-19 and beyond, where startups can seize the opportunity for social benefit by collaborating with other startups and with large corporations.

The following first-order themes appeared as commercial opportunities for SE during the time of disasters and pandemics. There are few studies on these themes, but they cover and discuss the important and different ideas and issues. These themes include localized and feasible solutions to global issues, lifestyle entrepreneurs, and SE in commercial/conventional entrepreneurship.

Studies in this field emphasized that the arenas of SE are not limited; they can address the issues of global scale with a localized answer. Especially in the unbridled times of social distancing, shaky stock markets, and lockdown of commercial operations across the globe, history has proven that mankind can overcome these difficult times, and entrepreneurs can lead us on this front [130].

However, today, poverty, infant mortality, lack of access to basic healthcare, proper sanitation, and clean drinking water, are all local and regional issues with a global impact. SE can help the society and also the global community to address and resolve these matters, first, and most importantly, by increasing international collaboration, which can create collaborative opportunities based on globalization and access to greater resources for

addressing the major problems with localized approaches, and local problems with global approaches. Second, these problems can also be addressed with the help of the available technology and resources, which can catalyze the solutions to global problems, including pandemics [131].

Makhlouf states that although SE start and continue to work within local geographical boundaries most of the time, their impact can be felt globally, and they are considered the agents of change without borders [132]. The business philosophy of SE revolves around societal and environmental issues with economic well-being, and this is different from their conventional/commercial counterparts, which makes them modern heroes. They are also leaders because they accept the challenge and do not unnerve their competitors and impersonators. Most of the time, they show how to pursue the same road to make an impact.

Another interesting and rare first-order theme that appeared is lifestyle entrepreneurship. These entrepreneurs are the individuals who first decide about the lifestyle they enjoy the most and then develop their business in a manner that suits them. This is completely different from other entrepreneurs, who prioritize the business and change the lifestyle according to work. Lifestyle entrepreneurs maintain their freedom and passion, they are less concerned about growth and maximization of profit, and rather, they work to support their lifestyle. They work for their passion with an idea for personal reward [133]. Though the concept of lifestyle entrepreneurs is relatively new, many entrepreneurs from the past can be tagged as lifestyle entrepreneurs [134].

Lifestyle entrepreneurs can come up with innovative ideas to cope with the COVID-19 pandemic, much like their startups, which are mostly based on the internet and smartphones. The idea of lifestyle entrepreneur/ship completely adheres to the guidelines of social distancing and can also motivate many other people to follow them. All they need to do is follow their dream lifestyle with a passion for social uplift and economic well-being, and, in these times of global unrest, the world needs them. There are many ways to start lifestyle entrepreneurship with feasible ideas and plans [135,136].

Beyond the above, another very important theme that appeared is economic depression, which creates greater opportunities for SE in time of economic downturns. Economic depression creates financial, as well as social, unrest, not only for individuals but also for organizations; yet, some studies revealed that financial unrest brings unique opportunities for SE to initiate and grow. The reason for this was further discussed by Cox; in his article, he explained that business models that are built on ethical grounds and supply goods and services with the aim of social betterment work better than models based on mere profit maximization and exploitation [137,138].

Globally, COVID-19 has created havoc. Analysts are not confident about the health of the economy, and they worry about another depression. Airlines are not flying much. Business is postponing their next projects and launches, and SE are asking how to survive during uncertainties. Yet, SE is able to treat this time as an opportunity. The basic difference between the commercial and SE is the approach. For SE, global economic instability does not pose a threat, but it is an opportunity [139].

Petrella and Richez-Battesti warned that if SE quits the basic idea of participation and democratic governance with dynamic stakeholders and resources and the foundation, then it will not be able to separate its very existence from the commercial counterparts [140].

## 6. Conclusions and Recommendations

Social and environmental problems are ubiquitous around the world. Hence, politicians, business leaders, and members of society call on efforts that focus on social and environmental objectives. Some of these are pursued by governments and by semi-public organizations. However, there is no clear boundary concerning which social and environmental problems should be the responsibility of governments and which problems may, at least partly, be left open for the market, for private and other non-governmental organizations. To put it concisely, the gap between the social demand for essential services

and their supply through formal and informal institutions provides an opportunity for SE to fill the void.

This paper aimed to provide state-of-the-art systematic review of the SE literature within the context of disasters and pandemics in general, and COVID-19 in particular. Based on keyword searches on databases, such as Elsevier, Emerald, JSTOR, Springer, Scopus, ScienceDirect, and Google Scholar, 103 journal and magazine articles and reports were shortlisted, and content analysis was conducted through NVivo tool, and the themes that emerged through the process included commercial, government, and social development. On the basis of recurring references in the essential themes, this research found that teleconsultation and telesupervision, telehealth, and industry 4.0 hold a promising future during and post-COVID scenario, provided the SE focus on technology and social integration to narrow the gap in quality mental health, education, employment, and manufacturing services in low-resource settings, which are more severely impacted by the pandemic. The authors recommend that the scope of social entrepreneurial training should be widened to improve the relevant skills and expertise among the current and prospective SEs. Further, it is imperative that the governments, particularly from the developing countries, focus on the science, technology, engineering, and mathematics (STEM)-related education, which will equip the youth with entrepreneurial mindset and skills that are important in enhancing their innovativeness and resourcefulness.

The contribution of this research is two-fold. First, we covered the pandemic- and disaster-related SE literature with relevant real-life examples from across the globe, including the developed and developing country context. The second contribution is the incorporation of a mind-mapping tool to trace the entire scenario of SE in the time of social distancing in terms of the opportunities that were availed by entrepreneurs in all fields and walks of life. The projects and ventures offering a wide array of products and services can bring inspiration to social entrepreneurial minds and motivate aspirants to tread a similar path to successfully create value for the public living in the throes of threat and uncertainty of current times.

Another practical contribution is toward entrepreneurs looking to start and develop a venture under disaster and pandemic conditions. Blessed with an entrepreneurial mindset and ability to balance a dual and sometimes contradictory purpose, this research provides evidence of how SEs can creatively and innovatively combine and/create resources oriented toward a social goal. It highlights the myriad of ways in which the potential SEs can capitalize on the altered business modus operandi and, therefore, enhance its resilience and survival in uncertain times, including, but not limited to, COVID-19.

## 7. Research Implications

This research is a response to the call for highlighting the coordinated action of SEs. This concerted feat can be achieved by agilely orchestrating innovative arrangements to meet the society's basic and advanced needs while simultaneously acting "as a glue" holding together cross-sectoral solutions, as emphasized by Bacq and Lumpkin, (2020) [8]. The collection of narratives presented in the form of gleaned themes in this study will help the social entrepreneurs in scanning their environment and identifying the opportunities in commercial, government, and social arena hidden by the constraints of social distancing. In this way, this research builds a case for mobilization and participation in social change by creating awareness and inspiration regarding innovative ideas social entrepreneurs are utilizing to transcend the pandemic-related constraints and hurdles in product and service delivery [140–144].

Moreover, it guides the SE scholars to study social entrepreneurship from more pertinent theoretical perspectives, such as Schumpeterian and Kirznerian theory, which is aligned with social capital's aim of value creation in their local context (Shockley & Frank, 2011) and the theory of resource mobilization and participation. Mobilization of constituents [16], including society and resources, refers to the support for a social cause offered by the public for a social cause. In this manner, the social entrepreneur and his or

her venture is a catalyst for coordinated support of the social cause or mission and provides a benefit in terms of ecology, economy, and society. So, in the context of mobilization, the social entrepreneurs facilitate the promotion of a specific social cause by winning support from the public. However, this theory does not take into account the constraints exerted by uncertain times and pandemics while discussing the phenomenon of mobilization. The consequences of these uncertain times include lockdowns, emergencies, and physical distancing, which might affect the "mobilization" process. Therefore, this study contributes to this theory by identifying a perspective of innovative and decentralized problem solving through technology, such as teleconsultation and telesupervision, telehealth, and industry 4.0, among others, which can alleviate the mobilization process of social entrepreneurs. Hence, the identification of the "role of technology" and the context of "uncertain times" were missing from the mobilization theory, which is provided via this research [145–149].

Similarly, Schumpeterian and Kirznerian theory discusses the creativity and alertness of entrepreneurs in opportunity identification and the universality of social entrepreneurial behavior. They also suggest that social entrepreneurship produces tangible social effects, which are measurable. However, this study makes a contribution, by shedding light on the measurable social change led by entrepreneurial activities during, as well as post, the disasters, pandemics, and crises in low-resource settings.

## 8. Research Limitations, De-Limitations, and Future Research

The nature of this research is qualitative, which has led to a two-fold limitation. First, the study does not contribute to the operationalization of the social entrepreneurship construct. Secondly, the findings regarding SE's role in social value creation cannot be generalized to other contexts spanning different societies and cultures. However, this limitation is partially compensated by the fact that the study performs content analysis on the SE from both developed and developing countries, which means that it offers inspiration and guidance to social entrepreneurs, irrespective of their economic context. This confirms the notion of resourcefulness of social entrepreneurs, since through their innovativeness they are able to allocate scarce resources in an efficient manner.

**Author Contributions:** Conceptualization, S.M.K.; methodology, S.M.K.; software, S.M.K.; validation, M.K.K., M.M.Q.A.; formal analysis, S.M.K., M.K.K. and A.A.N.; investigation, S.M.K., M.K.K. and M.H.; resources, M.M.Q.A., A.A.N. and M.H.; data curation, S.M.K., M.K.K.; writing—original draft preparation, S.M.K., M.K.K.; writing—review and editing, S.M.K., M.K.K. and M.M.Q.A.; supervision, M.M.Q.A. and A.A.N.; project administration, M.H.; funding acquisition, A.A.N. All authors have read and agreed to the published version of the manuscript.

**Funding:** Researchers Supporting Project number (RSP-2022/87), King Saud University, Riyadh, Saudi Arabia.

**Institutional Review Board Statement:** Not applicable.

**Informed Consent Statement:** Not applicable.

**Data Availability Statement:** Data are publicly available in shape of research studies and articles and references.

**Acknowledgments:** Researchers Supporting Project number (RSP-2022/87), King Saud University, Riyadh, Saudi Arabia.

**Conflicts of Interest:** No conflict of interest. The funders had no role in the design of the study; in the collection, analyses, or interpretation of data; in the writing of the manuscript, or in the decision to publish the results.

## Appendix A. Methodological Procedures for Search, Selection, and Exclusion

A.   Criteria for defining social entrepreneurship and delineating its parameters:

   i.   Explicitly from the work of Mair and Martí (2006)

      a.   Social entrepreneurial response to disasters and pandemics;

      ii.     Peer-reviewed journal articles;

      iii.    Empirical AND conceptual review;

      iv.    Magazine and newspaper articles;

      v.     Reports;

      vi.    Editorials.

B.    Exclusion criteria on the basis of theoretical applicability:

      i.      Studies in which primary focus is not social entrepreneurial opportunity in pandemics and disasters;

      ii.     Studies focused on social enterprises and not the social entrepreneurships;

      iii.    Research published in edited books and conference proceedings;

      iv.    Studies discussing entrepreneurial education or research techniques for social entrepreneurship;

      v.     Research not electronically available or by other reasonable means.

C.    Search Method and Scope-Stage–I:

      i.      Full search of studies in academic journals related to the field from 2010 to 2020;

      ii.     Elsevier, Wiley Online Library Oxford Academic Journals, JAMA network, and Emerald Insight databases were used with general keywords;

      iii.    Sciencedirect search engine;

      iv.    Inclusion scale by general keyword search using Google Scholar and Google search engines;

      v.     Initial focus on abstract and title.

      vi.    Keywords:

            a.    Social Entrepreneur;

            b.    Entrepreneur and social;

            c.    Opportunities during and after disaster/crisis;

            d.    Opportunities during and after pandemics;

            e.    Response pandemic;

            f.    Importance of social entrepreneurs in post-disasters;

            g.    Social entrepreneurship in Pakistan.

      vii.   A focused search of key journals in the field, to make sure that related articles not using keywords were included. Focused search in:

            a.    *Journal of Social Entrepreneurship;*

            b.    *Journal of Global Entrepreneurship Research;*

            c.    *Entrepreneurship Theory and Practice;*

            d.    *Journal of Developmental Entrepreneurship;*

            e.    *International Journal of Social Entrepreneurship and Innovation;*

            f.    *Journal of Business Venturing;*

            g.    *Business Horizon.*

D.    Search method and scope-Stage II:

      i.      Manual reading/checking by senior investigators of all papers included in the database up to this point to include or exclude, based on fit with definitional and search parameters;

      ii.     Pattern-matching of Stage I list relative to published reviews by senior investigators.

E.    Search method and scope-Stage III:

      i.      Re-assess articles excluded by review but included elsewhere (and include where deemed appropriate).

## Appendix B. Procedures for Thematic Analysis and Ontological Organization

A.    Data organization:

       i.        Arranged papers/reports in chronological order from 2010 to 2020;

       ii.       Used NVivo for coding by researchers.

   B.    Theme Identification and Coding:

       i.        Researchers individually review articles and reports for ideas, getting the purpose of study, research problem, main arguments, methodology, and main hypotheses to define the aim of the study;

       ii.       An anecdotic statement highlighting the core idea is assigned to each study to find its conceptual terminology and vocabulary;

       iii.     After reading the studies, researchers stand the test between statements and solve the differences through discussion;

       iv.     Thematic names were assigned to each study, derived from anecdotic statements.

   C.    Ontological Organization:

       i.        Two thematic names for each study were assigned, after discussion, resulting in the first theme becoming first order (T1) for each paper;

       ii.       Themes were reviewed for duplication or repetition.

   D.    Thematic and ontological interpretation and validation:

       i.        Descriptors and themes are summarized in tables and sorted in chronological order by thematic area (super-theme), followed by second-order and first-order themes and corresponding descriptive statements as a domain ontology;

       ii.       The thematic structure of the field is mapped (Figure 1) and compared for consistency with Table 1;

       iii.     An interpretive account of each theme is written by returning to the papers, pattern matching against theme descriptors and ontological fit.

   E.    Quality Checking:

       i.        Every study (article, report) was equally treated and coded independently by researchers;

       ii.       Review process was systematic, in-depth, and comprehensive;

       iii.     Thematic map and ontology tables were developed to check the iteration and consistency;

       iv.     Themes were checked and rechecked against each other and original dataset;

       v.      These were tested for consistency and uniqueness;

       vi.     Datasets were interpreted for sense and common terms and expressions preserved;

       vii.    Themes were iteratively pattern matched with the data, and the ontology tables and thematic map were checked for consistency;

       viii.   Role of authors of each study is fully acknowledged.

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
