# Peer review of "Social Entrepreneurship Opportunities via Distant Socialization and Social Value Creation"

_sustainability, doi:10.3390/su14063170_

Round 1
Reviewer 1 Report
Dear Authors
when I started reading your work, it seemed to me appropriate, social entrepreneurship in times of covid, industry 4.0 among the keywords.
The literature review is somewhat sparse in terms of references, and the introduction without a very strong theoretical contribution but remediable to some extent.
Nothing indicated that the work was a literature review. The reader expects an empirical work in line with the above.
Without an empirical part and without a strong theoretical contribution the paper does not hold up, it is not in keeping with a journal with this impact index.
Author Response
Authors have incorporated almost every comment and suggestions by the respectable reviewer
- In the introduction, the background and necessity of the study are logically written.
- In the review of literature, the definitions concepts and theories of SE are discussed.
- In the conclusion section, the importance, contributions, and implications of the research findings are elaborated.
- 4. In Appendix A.(dataset), the list of journals and magazines used in this paper should be displayed. (author, title, year, IF, main theme, etc.)
- Corrections are made to remove unnecessary capital letters and spelling mistakes.

Reviewer 2 Report
Your article has an interesting subject. It discusses the social entrepreneurship opportunities via distant socialization and social value creation.
However, this article lacks logical development. The implications for social entrepreneurs through literature review are also insufficient.
[1] Introduction
In the introduction, the background and necessity of the study should be logically written.
[2] Literature review
2.1. Theoretical underpinnings of Social Entrepreneurship
This study needs an academic concept and arrangement on social entrepreneurship. For example, it is necessary to explain the definitions, concepts, and components of social entrepreneurship.
[3] Conclusions
Authors must elaborate more on what is their contribution to the literature as well as an opportunity for future research. Questions that you need to answer:
(1) Why is your study important?
(2) How it could extend to the existing knowledge on the issue/topic? I suggest you concentrate on the description of the implications of your main findings
[4] Appendix A.(dataset)
The list of journals and magazines used in this paper should be displayed.(author, title, year, IF, main theme, etc.)
Please correct it.
p.2.(61) ; SE --> social entrepreneurship(SE)
p.7.(297): requirements. --> requirements
p.2(86): Entrepreneurs --> entrepreneurs
p.3(143): Pneumonia --> pneumonia
p.4(183): Organizational Problem --> organizational problem
p.5(200): Resilience --> resilience
p.5(218): 3.1 . --> 3.1.
p.6.(297): Opportunity to Assist Government --> opportunity to assist government
p.6(298)(299): Addressing ; Commercial Opportunities
p.7.(305): Opportunities
p.8.(312): Opportunities
p.9.(318): Entrepreneurial Opportunityes ?
p.14.(579): Personal and Protective Equipment(PPE)
p.14.(613): Agriculture --> agriculture
p.15.(639): Covie-19 -->COVID-19
p.15.(662): These themes include Localized 662 and feasible Solutions of global issues, Life Style Entrepreneurs, and SE In Commercial/conventional entrepreneurship out.
p.18.(756): 1. ppendix B --> Appendix B
Author Response
Authors have incorporated almost every possible changes suggested by the reviewer;
- In the complete manuscript, the suggested referenced style (MDPI) is applied
- Required changes to the English language and style are made.
- The limitations and future directions of research are included in the conclusion.
- Research Methodology and Data and Analysis are segregated and explained.

Reviewer 3 Report
Please check the attachment.

Author Response
Authors have incorporated following suggestions in the manuscript;
- In the complete manuscript, the suggested referenced style (MDPI) is applied.
- The paper provides clearly indicated main aim and research objectives.
- Strong emphasis is laid on COVID-19 pandemic and SE during this pandemic. However, examples and cases are referred from pre-COVID era in order to discern a logical pattern regarding the pattern of SE’s resourcefulness in past disasters and pandemics.
- Recommendations are elaborated.
- The limitations and future directions of research are included in the conclusion.

Reviewer 4 Report
Thank you for the opportunity to review this paper. The paper is looking into rather interesting topic, however, there are several comments to address: • Throughout the paper, where authors mention contributions of other authors, they should always indicate the number of the reference in the brackets (e.g. “Similarly, according to Drucker (1985) entrepreneur is a person who is always in search of a change, responds to opportunities, and exploits it accordingly.” – number of reference needs to be added [14]). • The paper lacks clearly indicated main research question. • The major issue of the paper is that the aim of the paper, abstract, introduction and some parts of the discussion and conclusion are strongly emphasizing COVID-19 pandemic and SE during this pandemic. This is not aligned with the period of the analysis starting in 2010…The main focus of the paper should be re-examined, therefore, the several parts of the paper should be somewhat rewritten and reorganized, or the analysis should be re-done according the focus on COVID-19 pandemic. • Authors should elaborate their recommendations: e.g. “The authors recommend that the scope of social entrepreneurial training should be widened, to improve relevant skills and expertise.” This is a rather general statement, without any elaboration, suggestions or critical reflection. • The concluding part of the paper lacks the elaboration of the limitations of the study.

Author Response

(The authors gave the same response as above.)

Round 2
Reviewer 1 Report
Dear Authors,
you did not answer my review one.
You have improved the manuscript however, only 4-5 studies that you cited are about COVID 19.
Sentences as:
"Studies emphasize that social entrepreneur’s role in post-disaster/pandemic recovery 463
must be appreciated and recognized because they hold an important position in the as- 464
sessment of rapidly changing situations."
These sentences are stronger that the analysis that you have performed.
Research theoretical contribution is still weak.
Industry 4.0 is not needed in the keywords, because you do not write too much about that concept.
Minor point, some references are not in the format of the journal template.
Author Response
Thank you for sharing the expert comments and suggestions on the manuscript titled, “Social Entrepreneurship Opportunities via Distant Socialization and Social Value Creation.”
We have tried our best to meticulously incorporate and respond to the points raised by the Reviewer-02. Please note that grammatical errors are removed and the document is proofread by the language expert. Further, all changes are made by using the ‘track changes’ function for the convenience of the reviewers. In the track changes mode, reviewers can see that their comments/suggestions were incorporated and commented.
In the attached file you can see that your suggestions and reviews were duly incorporated.

Reviewer 4 Report
Thank you for your effort to address all the comments and to revise the paper.
Author Response
Dear Reviewer,
We the authors really appreciate, your time and effort for making this study better and giving your suggestions for a contribution in the field.
Regards
Round 3
Reviewer 1 Report
Welldone. Congratulations